# Effect of Nano-Sized TiC-TiB₂ on Microstructure and Properties of Twin-Roll Cast Al-Cu-Mn-Zr Alloy

**Jiaming Cao [1], Bao Wang [1], Xiao Liu [1], Ting Chang [1] and Qinglong Zhao [1,2,*]**

[1] Key Laboratory of Automobile Materials, Ministry of Education, School of Materials Science and Engineering, Jilin University, No. 5988 Renmin Street, Changchun 130025, China; caojm19@mails.jlu.edu.cn (J.C.); wangbao20@mails.jlu.edu.cn (B.W.); xiaoliu19@mails.jlu.edu.cn (X.L.); changting19@mails.jlu.edu.cn (T.C.)

[2] State Key Laboratory of Automotive Simulation and Control, Jilin University, Changchun 130025, China

[*] Correspondence: zhaoqinglong@jlu.edu.cn

**Abstract:** Al-5Cu-0.8Mn-0.1Zr strips produced during a twin-roll-casting process are investigated in this paper. Central segregation does occur in the alloy strip, and the segregation is a mixture of $Al_2Cu$ and $Al_{20}Cu_2Mn_3$ eutectic phases. This mixture is difficult to be dissolved in solid solution heat treatment. In this paper, nano-sized TiC/TiB₂ particles are introduced in the process of twin-roll-casting. It was found that after adding 0.3% nano-sized TiC-TiB₂ particles, the composition distribution becomes more uniform, and the central segregation is eliminated, compared to the strip without nanoparticles. The tensile strength of T6-treated alloy increases from 350 MPa to 390 MPa when nanoparticles are added, and the goal of increasing the properties of twin-roll-cast Al-Cu-Mn-Zr alloy has been achieved.

**Keywords:** twin-roll-casting; Al-Cu-Mn-Zr; nanoparticles; central segregation; composition homogenization; solidification

## 1. Introduction

Twin-roll-casting (TRC) is a technology with an extremely high cooling rate, which combines casting and rolling processes [1,2], so it can form a supersaturated structure and significantly improve the solid solubility of elements in the material [3]. Because the apparent central segregation in the material caused by the TRC process is challenging to eliminate [4,5], the mechanical properties of the material will be lower than conventional casting.

Al-Cu alloy is widely used in the automobile and aircraft industry [6,7]. The application of the TRC process can significantly shorten the production time of the Al-Cu alloy sheet and improve the production efficiency, so that Al-Cu strips can be attained more quickly. It can also produce special Al-Cu alloys with better heat high-temperature properties. The strength and hardness of commonly used Al-Cu alloys will decrease if the temperature is above 523 K. Above this temperature, the θ' phase changes to a steady θ Phase [8,9]. To improve the high-temperature stability of Al-Cu alloy, Mn and Zr are introduced into the alloy. The purpose of adding Mn and Zr is to produce T phase and $Al_3Zr$ with better thermal stability. It can not only improve the room temperature performance, but also improve the high-temperature stability of the material. Therefore, the mechanical properties of Al-Cu-Mn-Zn (ACMZ) can be slightly enhanced at high temperatures [10]. However, the element enrichment area produced in the TRC process will lead to the accumulation of a large number of T phases and $Al_2Cu$ in the center, which is named central segregation. The central segregation cannot be eliminated in solution treatment and will affect the uniformity of internal components of the alloy and reduce the high-temperature properties.

The grain refinement of the Al-Fe-Si foil strip was realized by introducing a kind of grain refiner whose main component is TiC-TiB$_2$ [11]. The central segregation of Al-Cu alloy was significantly decreased by unifying the components, and the grain refinement was reduced through the joint action of nano-sized TiC and the TRC processes [12]. They also made the grains finer by cold rolling before heat treatment, which also improved the mechanical properties of the alloy. The properties of rolling Al-Mg-Si alloy with different nanoparticles (TiC, TiB$_2$, TiC-TiB$_2$) were compared. Finally, it was found that the properties of Al-Mg-Si alloy using TiC-TiB$_2$ master alloy as an additive were best [13]. In addition to homogenizing the composition, nanoparticles can also improve the strength, hardness, and high-temperature properties of materials [14,15]. Previous experiments have proved that the introduction of plastic deformation can not only refine the grains in the subsequent heat treatment, but also improve the distribution of nanoparticles and reduce the accumulation of particles [16]. Therefore, in order to obtain TRC ACMZ strips with uniform composition and refined grains, nano-sized TiC-TiB$_2$ and a cold rolling process are introduced into the TRC process.

## 2. Materials and Methods

An asymmetric TRC machine made by Gonghui metallurgical equipment technology (Suzhou) Co., Ltd. (Suzhou, China) was used in this study, as shown in Figure 1a. Al-5Cu-0.8Mn-0.1Zr was prepared by using pure aluminum, copper, and master alloy composed of Al-10Mn and Al-10Zr. Then, the alloy was cut into small pieces of about 120 g each. This is the optimal weight to pass through the rollers and form a shaped strip of appropriate size before solidification. The rolling gap is 0.5 mm, and the rolling speed is 10 m/min. The re-melted and fully stirred alloy sheets are prepared into TRC alloy strips with a thickness of 2–3 mm through the TRC process. Excessive thickness will lead to an increase in rolled plate defects, and insufficient thickness will not allow polishing. The master alloy containing 30 wt.% TiC-TiB$_2$ nanoparticles is added in the re-melting process. Among them, the ratio of TiB$_2$ to TiC is 2:1, the size of TiB$_2$ is 450–700 nm, and the size of TiC is between 150–230 nm. The final composition of the alloy strip is Al-5Cu-0.8Mn-0.1Zr-x TiC-TiB$_2$ (x = 0, 0.1, 0.3). In addition, Al-5Cu strips and Al-5Cu-0.3Mn-0.1Zr strips were prepared during the TRC process in comparison with as-cast microstructure. The roller is made of copper with no lubrication on the surface. As shown in Figure 1c, the ACMZ strip is rolled up to a reduction in thickness of 50%, then isothermal holding for 6 h at 673 K and cooling in air. The purpose of this step is to precipitate dispersed T phase with high thermal stability after deformation, because the T phase is mainly formed and grown during homogenization heat treatment [10,17]. Subsequently, the strip is heated to 848 K for 5 h and artificially aged at 438 K for 6 h (T6). It is quenched in water after solution treatment and cooled in air after T6 heat treatment. This process parameter refers to the experiments of others and is adjusted for the existing equipment [18,19]. The ACMZ thin strip after heat treatment is cut into 13 mm ×13 mm samples and divided into three batches to make room-temperature hardness samples, scanningelectron microscope (SEM) samples, and opticalmicroscopy (OM) samples. The cross-sections of SEM samples and OM samples were mechanically polished. Then, the OM samples were anodized with a 5% HBF$_4$ solution. The surfaces on both sides of the hardness sample were mechanically polished, and then the surface hardness was tested with a microhardness tester from Laizhou Huayin company (Laizhou, China). Microhardness values of 10 points were collected on each surface. The gauge distance of the tensile specimen was 10 mm, the thickness was about 1.5 mm, and the width was about 4.5 mm. The surface was mechanically polished, and the movement rate of the collet was 0.6 m/min.

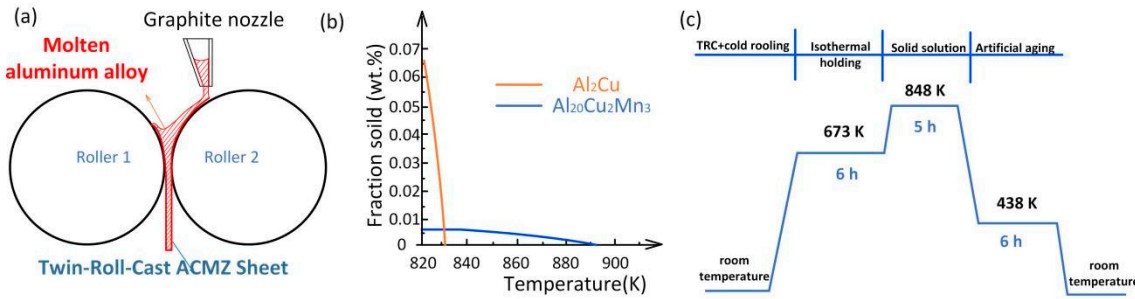

**Figure 1.** (**a**): Solidification process of asymmetric TRC machine; (**b**): Phase content after solidification calculated by JMatPro according to the Scheil equation; (**c**): Heat treatment process for all alloys.

## 3. Results and Discussion

In the solidification process of this TRC process, due to significant temperature difference between the two surfaces during solidification, there will be considerable differences in microstructure and structure at various positions in the material. To judge the composition of the primary phase, phase content after solidification is calculated by JMatPro-v11.2 according to the Scheil equation as a reference. The Scheil equation is used to describe the solute redistribution at the solid–liquid interface under non-equilibrium solidification conditions, which is different from the actual situation of the TRC process, so it is only used to judge the composition of the primary phase.

### 3.1. Microstructure

Figure 2a shows that the microstructures on both sides are significantly different between the central segregation region. According to the EDS results of Figure 3, and combined with Figure 1b, we can infer that the primary phase is a mixture of T phase and $Al_2Cu$ phase. From Figure 2b,c, a similar primary phase can be found on both surfaces. Figure 2d–f shows primary phases with different shapes close to the preferentially solidified surface, near the central segregation area, and the later solidified surface. As can be observed, the primary phases close to the preferentially solidified surface are grid-like. There is a transition structure from the grid to lamellar in the central region, and the other side is lamellar. It shows that the asymmetric TRC process leads to one side of the alloy solidifying first, where the cooling rate is fast. After the grid primary phases are formed, the solid–liquid interface is pushed forward. On the other side, the non-solidified aluminum liquid contacts the roller and forms an aluminum alloy shell on the surface. However, the solidification temperature is much lower than the priority solidification side. The solidified shells on both sides rotate with the roller and are combined at the kiss point to form an element-enriched area. After solidification, it is the central segregation area. The solidification temperature of the first solidified side is high, and the undercooling is more remarkable, so a grid primary phase is formed. On the other side of the central segregation region, the Cu content increases, and the undercooling is limited; thus, a sizeable lamellar primary phase is formed.

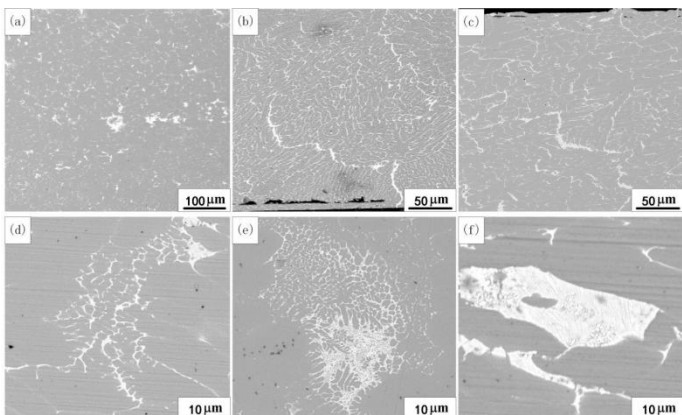

**Figure 2.** The primary phases of as-cast ACMZ. (**a**) The central segregation region; (**b**) The preferentially solidified surface; (**c**) The later solidified surface; (**d**) Primary phases near the preferentially solidified surface; (**e**) Primary phases between the central segregation region and the preferentially solidified surface; (**f**) Primary phases near the later solidified surface.

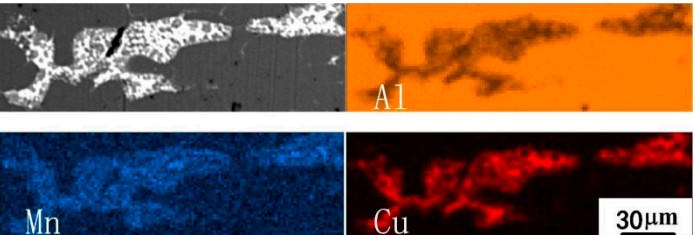

**Figure 3.** The EDS data of the primary phase in the central segregation region of ACMZ, as cast.

Compared with Figure 4a–c, it can be found that with the increase in nanoparticle content, the segregation area in the as-cast structure of the alloy gradually decreases, and the large pieces of primary phases also disappear. The structures on both sides of the central segregation area tend to be consistent. It demonstrates that by adding nanoparticles, the composition uniformity in the alloy is improved. Comparing with Figure 4e–g, it can be found that ACMZ and ACMZ-0.1 nanoparticles will result in large residual second phases after isothermal holding + solid solution treatment (IS), and these undissolved residual areas roughly maintain the contour of the original large $Al_2Cu$, as shown in the small figure of Figure 4e. The T phase is an insoluble phase. After solution treatment, the shape of the T phase aggregation area is approximately the same as that of the previous sheet θ. According to Figure 5, this area is the mixture of $Al_2Cu$ and T phase. It shows that the T phase precipitates on $Al_2Cu$, and it hinders the dissolution process of $Al_2Cu$ during solution. In ACMZ-0.3 nanoparticles, because the as-cast microstructure is uniform and no large segregation area exists, only a tiny amount of residual T phase appears at the edge of the grain boundary after isothermal holding and solution treatment. After isothermal holding, dispersed T phase with a diameter of about 200–400 nm precipitates in the alloy, and the content of T phase in $\alpha$(Al) will increase slightly with the increase of nanoparticle content compared with Figure 4d,f.

Figure 6 and Table 1 show the solute atom content in $\alpha$(Al) at different positions of as-cast ACMZ and ACMZ-0.1 nanoparticles. Because the value obtained by point scanning has great fluctuation, we chose the result of linescan, which shows the average value of element content on the line. The linescan data were collected from the central segregation region and surface part of the two alloys, and the results are shown in Table 1. Comparing with Figure 5a–d, it can be found that by adding nanoparticles, the content of solute atoms decreases in the central segregation region and increases on the surface.

Based on the above experimental results, it can be found that the addition of nanoparticles effectively homogenized the alloy composition, reduced the central segregation,

and promoted the dispersion and precipitation of T phase. This is because the introduction of nanoparticles increases the solidification nucleation sites, promotes the formation of more equiaxed grains in the middle of the alloy and reduces the formation of columnar grains [13]. Therefore, solute atoms cannot reach the central segregation region along the columnar grains, to make the alloy composition in the alloy more uniform. After the addition of 0.3 wt.% nanoparticles, the dispersed T phases in the alloy increases significantly because the 0.3 wt.% nanoparticles eliminate the central segregation region, make the distribution of Mn element in the alloy more uniform, and increase the content of Mn in other regions. Therefore, more T phases can be precipitated in the aluminum matrix.

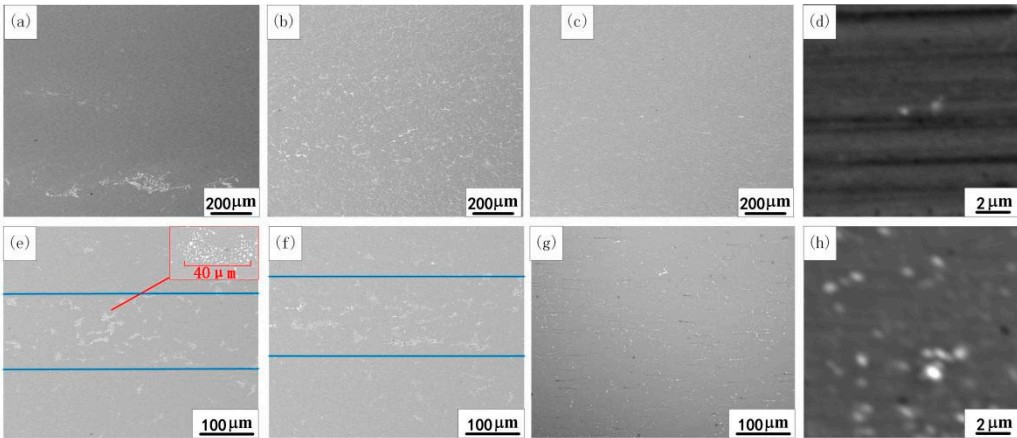

**Figure 4.** SEM results of ACMZ alloy with different content of nanoparticles after as-cast and isothermal holding + solid solution (IS) treatment. (**a**) ACMZ, as-cast; (**b**) ACMZ-0.1 nanoparticles, as-cast; (**c**) ACMZ-0.3 nanoparticles, as cast; (**d**) T phases in ACMZ, isothermal holding; (**e**) ACMZ, IS; (**f**) ACMZ-0.1 nanoparticles, IS; (**g**) ACMZ-0.3 nanoparticles, IS; (**h**) T phases in ACMZ-0.3 nanoparticles, isothermal holding.

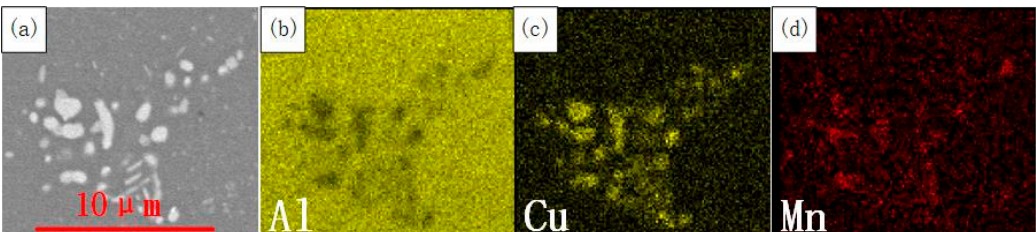

**Figure 5.** The EDS map of the residual part after solid solution, ACMZ. (**a**) SEM area of residual structure after solid solution; (**b**) Al in this area; (**c**) Cu in this area; (**d**) Mn in this area.

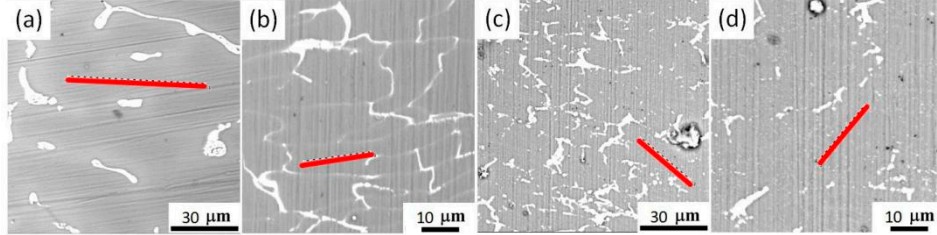

**Figure 6.** Scanning areas of linescan in $\alpha$(Al) at different positions of as-cast ACMZ and ACMZ-0.1 nanoparticles, and the red lines are the paths of line scanning. (**a**) The central segregation region of ACMZ; (**b**) The surface of ACMZ; (**c**) The central segregation region of nanoparticles; (**d**) The surface of ACMZ-0.1 nanoparticles.

**Table 1.** The results of linescan in $\alpha$(Al) at different positions of as-cast ACMZ and ACMZ-0.1 nanoparticles. Line (a): The central segregation region of ACMZ; Line (b): The surface of ACMZ; Line (c): The central segregation region of nanoparticles; Line (d): The surface of ACMZ-0.1 nanoparticles.

| Element Content (wt.%) | Line (a) | Line (b) | Line (c) | Line (d) |
|---|---|---|---|---|
| Cu (wt.%) | 2.29 | 1.24 | 2.48 | 1.73 |
| Mn (wt.%) | 0.78 | 0.60 | 0.71 | 0.63 |

Figure 7a,b shows significant differences in grain size and morphology between cold-rolled ACMZ and ACMZ + 0.3TiC-TiB$_2$ alloys. More refined grains with more homogeneous distribution can be observed in the latter alloy due to the increased amount of added nanoparticles. Also, prominent shear bands and grain breakage were detected in the ACMZ alloy, while in the latter one, these phenomena were only minimally observed. These results demonstrate that the addition (introduction) of nanoparticles can effectively refine the grain size during solidification, demonstrating that nanoparticles provide more nucleation sites for heterogeneous nucleation to form finer equiaxed structures in an ACMZ alloy. The microstructures after isothermal holding, as seen in Figure 7c,d, exhibited recrystallized structure for both alloys. However, the coarsening of grains in the ACMZ + 0.3TiC-TiB$_2$ alloy was less significant than that in the ACMZ alloy. In ACMZ + 0.3TiC-TiB$_2$, since nanoparticles provide more equiaxed nucleation sites and hinder the formation of columnar grains, the precipitation phases are more uniform, and the coarse second phase at the central segregation is reduced. Therefore, the uniformity of recrystallization structure is improved, and finer and more uniform grains are formed.

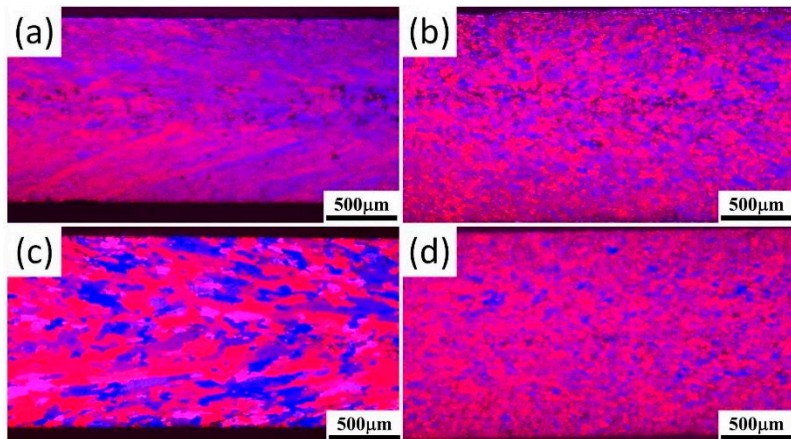

**Figure 7.** OM test results: (**a**) ACMZ, as cast; (**b**) ACMZ + 0.3TiC-TiB$_2$, as cas; (**c**) ACMZ, isothermal holding; (**d**) ACMZ + 0.3TiC-TiB$_2$, isothermal holding.

### 3.2. Mechanical Properties

In Table 2, it can be found that the hardness data of the two surfaces of the sheet metal are slightly different. After solution treatment, the hardness values of ACMZ + 0.3TiC/TiB$_2$ surfaces are significantly higher than those of the other two alloys. Combined with the results shown in Figure 4d,h, it can be demonstrating that the hardness of the material can be improved by the dispersion distribution of T phases. The research of Zhang et al. also confirmed that the strength of the material was significantly improved through the joint action of fine grain strengthening and the Orowan mechanism [20]. After T6 treatment, no matter which alloy, the increase in hardness value is about 40HV, indicating that the effect of nanoparticles on aging precipitates is not significant.

**Table 2.** Average value and standard deviation of surface hardness (HV) of different alloys under different heat treatment conditions.

| Heat Treatment | ACMZ | ACMZ + 0.1TiC-TiB$_2$ | ACMZ + 0.3TiC-TiB$_2$ |
| --- | --- | --- | --- |
| Solid solution | 83.0 ± 3.2 | 84.2 ± 2.9 | 91.9 ± 3.1 |
| Artificial aging | 126.5 ± 4.1 | 129.6 ± 4.2 | 134.1 ± 3.0 |

In Figure 8, it shows that after T6 heat treatment, due to adding nanoparticles, the yield strength increases slightly. The tensile strength of the blank sample is about 350 MPa, the tensile strength of ACMZ with 0.1 wt.% and 0.3 wt.% nanoparticles has improved, and the tensile strength with 0.3 wt.% particles has increased to 392 MPa. It can be considered that adding nanoparticles promotes the precipitation of dispersed T phases and improves the tensile strength. However, there are still many central segregations in ACMZ-0.1 nanoparticles, which reduces its plasticity.

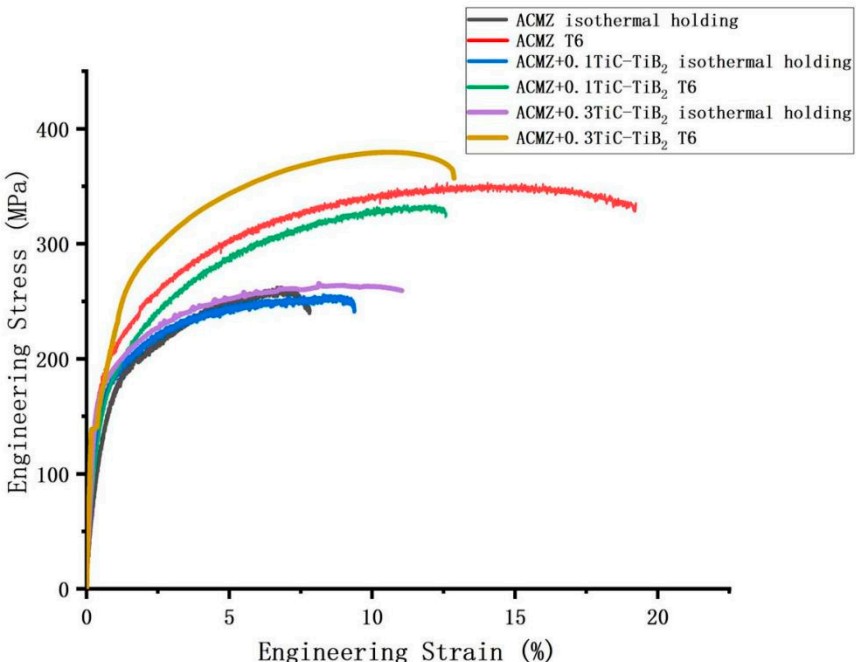

**Figure 8.** Stress–strain curves of alloys with different compositions.

### 4. Conclusions

This research shows how nanoparticles can improve the properties of ACMZ alloy.

(1) Adding nano-sized TiC-TiB$_2$ particles to an ACMZ alloy can make the structure and composition more uniform and reduce central segregation. By introducing nanoparticles, the grains are refined because it provides more nucleation sites for heterogeneous nucleation, and the columnar grains are reduced. Solute atoms cannot come to the central segregation region along the columnar grains and produce enrichment; therefore, in the TRC ACMZ strips with nanoparticles, the Mn content of aluminum matrix will be higher than ACMZ, so more dispersed T phases will be precipitated during isothermal holding. It can also make the recrystallization structure more uniform and the grains finer.

(2) After T6 heat treatment, the tensile strength of the ACMZ + 0.3 nanoparticles strip made by this method is better than ACMZ. Nanoparticles eliminate the central segregation, resulting in an increase in dispersed T phases, which improves the tensile strength of the alloy through dispersion strengthening.

**Author Contributions:** Conceptualization, Q.Z. and J.C.; methodology, Q.Z.; software, J.C.; validation, B.W., X.L., T.C. and Q.Z.; formal analysis, J.C.; investigation J.C.; resources, J.C.; data curation, J.C.; writing—original draft preparation, J.C.; writing—review and editing, J.C., B.W., X.L., T.C. and Q.Z.; visualization, J.C.; supervision, J.C.; project administration, Q.Z.; funding acquisition, Q.Z. All authors have read and agreed to the published version of the manuscript.

**Funding:** This work funded by the National Natural Science Foundation of China (Grant No. 51790483).

**Data Availability Statement:** The data used to support the findings of this study are available from the corresponding author upon request.

**Conflicts of Interest:** The authors declare no conflict of interest.

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
