# Peer review of "Effect of Nano-Sized TiC-TiB2 on Microstructure and Properties of Twin-Roll Cast Al-Cu-Mn-Zr Alloy"

_metals, doi:10.3390/met12040563_

Round 1

Reviewer 1 Report

Thank you very much for your comments. Now the manuscript has been improved and many points have become clearer. I recommend this manuscript after some minor revision.

- Phrase ‘primary second phases’ is confusing. Just ‘primary phases’ is more appropriate. Or worse variant ‘primary second-phases’.

- You write both ‘ACMZ-0.1% nanoparticles’ and ‘ACMZ-0.1 nanoparticles’. Use ‘%’ sign throughout the text. The same for ‘ACMZ-0.3% nanoparticles’ and ‘ACMZ-0.3 nanoparticles’. Carefully monitor the text of the manuscript is unified.

- Line 222. Use a capital letter in a word ‘conclusion’

- Table 2. Only one digit after decimal point is required.

Reviewer 2 Report

The manuscript is revised adequately according to the reviewers comments.

There is one question about the added text, which is related to the previous point no.6.

What is the Scheil's equation?

Author Response

This manuscript is a resubmission of an earlier submission. The following is a list of the peer review reports and author responses from that submission.

Round 1

Reviewer 1 Report

The authors investigated the effect of nanoparticles addition in microstructure modification of rolled-casting of an aluminium alloy. They eliminated the centre segregation during solidification and refined the grains which led to improvement in mechanical properties. The paper is written well and concisely. Before acceptance, following points need to be addressed:

  • What is the purpose of isothermal holding?
  • In figure 4 the caption lacks the explanation for figures 4 d and e.
  • Explain what the criteria are for choosing the process parameters.
  • Explain the mechanism of hinderance of T phase production by adding nanoparticles. It is important to provide this mechanism.

Author Response

Point 1: What is the purpose of isothermal holding?

Response 1: The purpose of this step is to precipitate dispersed T phase with high thermal stability after deformation. 

Point 2: In figure 4 the caption lacks the explanation for figures 4 d and e.

Response 2: This error has been corrected.

Point 3: Explain what the criteria are for choosing the process parameters.

Response 3: This process parameter refers to the experiments of others and is adjusted for the existing equipment.

Point 4: Explain the mechanism of hinderance of T phase production by adding nanoparticles. It is important to provide this mechanism.

Response 4: After discussion, we changed a more reasonable theory to explain this phenomenon. Because the yield strength of the alloy is very similar, it can be considered that the addition of nanoparticles hinders the movement of dislocations, increases the dislocation density, then improves the work hardening rate, and finally leads to the increase of tensile strength.

Reviewer 2 Report

This paper describes the usefulness of the twin-roll-casting for the uniform dispersion of  nano-particles (TiC/TiB2). Although the process brings preferable tensile properties as shown in Fig.6 (SS curves), too much lack of the information about experimental conditions. This means that this report cannot support the reproducibility of experimental results. The adequate reversion should be needed at least at the following points.

(1) p.1, L.18, abstract: '... became more uniform,...'

The target for comparison is not clearly described (the sample with 0wt% or 0.1% TiC-TiB2 particles?)?

(2)P.1, L.42, abbreviation without no definition.

'ACMZ' is not defined at the first appearance of this paper.

(3) P.2, L.~63. twin-roll-casting

The surface condition of the casting roller should be mentioned, such as lubricant, material of the roller etc, if it is possible.

(4) P.2, L.67-68. '.. the ACMZ strip is rolled with a rolling amount of 50%.'

The reason why the cast strip was cold-rolled should be mentioned. 'with a rolling amount of 50' should be 'up to a reduction in thickness of 50%.

(5) P.2, L.69. (aged for 6h at 673K)

Is it possible to refer the heat treatment condition of this study as T6? (The abbreviation, T6, is appeared at the later parts of this paper.)

(6) P.2, Fig.1(b) and P.2, L.80-86 (the first paragraph of the section 3)

The phase fraction as a function of temperature is presented with the calculation by JMatPro database. The preconditions for the calculation does not seem to reflect the real condition because the studied process is so special that it should be studied.

(7) P.5, Fig.6 and L.176-184.

The condition of the tensile test ( sample shape, deformation speed etc) is not found. In addition, the discussion about the comparison between the results of this work and the calculation with JMatPro database has no meaningful information since the tensile properties changes according to their microstructure.

(8) The reason for the higher tensile strength of the ACMZ-0.3TiC/TiB2, T6 compared with that of the ACMZ-0.1TiC/TiB2 should be described with mechanism other than the grain  refinement strengthening because the yield strength of these are similar to each other.

Author Response

Point 1: p.1, L.18, abstract: '... became more uniform,...'

The target for comparison is not clearly described (the sample with 0wt% or 0.1% TiC-TiB2 particles?)?

Response 1: We have made supplementary explanations on this issue. We compared the microstructure and properties of ACMZ alloys with different content of nanoparticles.

Point 2: P.1, L.42, abbreviation without no definition.

'ACMZ' is not defined at the first appearance of this paper.

Response 2: This error has been corrected.

Point 3: P.2, L.~63. twin-roll-casting. The surface condition of the casting roller should be mentioned, such as lubricant, material of the roller etc, if it is possible.

Response 3: The roller is made of copper with no lubrication on the surface.

Point 4: P.2, L.67-68. '.. the ACMZ strip is rolled with a rolling amount of 50%.'

The reason why the cast strip was cold-rolled should be mentioned. 'with a rolling amount of 50' should be 'up to a reduction in thickness of 50%.

Response 4: This error has been corrected.

Point 5: P.2, L.69. (aged for 6h at 673K). Is it possible to refer the heat treatment condition of this study as T6? (The abbreviation, T6, is appeared at the later parts of this paper.)

Response 5: This error has been corrected.

Point 6: 

P.2, Fig.1(b) and P.2, L.80-86 (the first paragraph of the section 3). The phase fraction as a function of temperature is presented with the calculation by JMatPro database. The preconditions for the calculation does not seem to reflect the real condition because the studied process is so special that it should be studied.

Response 6: The tensile strength of conventional casting is calculated by JMatPro software. The existing problem of TRC process is that the performance of the plate prepared by TRC process is far lower than that of conventional casting, so we compare the TRC plate prepared by us with that of conventional casting.

Point 7: 

P.5, Fig.6 and L.176-184.

The condition of the tensile test ( sample shape, deformation speed etc) is not found. In addition, the discussion about the comparison between the results of this work and the calculation with JMatPro database has no meaningful information since the tensile properties changes according to their microstructure.

Response 7: 

The gauge distance of the tensile specimen is 10mm, the thickness is about 1.5mm, and the width is about 4.5mm. The surface and sides are polished to 2000# sandpaper, and the tensile rate is 0.6mm/min.

Point 8: The reason for the higher tensile strength of the ACMZ-0.3TiC/TiB2, T6 compared with that of the ACMZ-0.1TiC/TiB2 should be described with mechanism other than the grain  refinement strengthening because the yield strength of these are similar to each other.

Response 8: 

Because the yield strength of the alloy is very similar, it can be considered that the addition of nanoparticles hinders the movement of dislocations, increases the dislocation density, then improves the work hardening rate, and finally leads to the increase of tensile strength.

Reviewer 3 Report

In this paper, the authors attempted to refine the grain and improve the uniformity of the chemical composition of the Al-Cu-Mn-Zr alloy by adding TiC - TiB2 nanosized particles. The topic of the article is interesting, however I consider that the following points are critical problem for publishing.

1) The article is submitted as an original scientific manuscript, but its length is similar to a short communication. Therefore, a much more detailed description of the methods and results/discussion is required.

2) Why are two types of nanosized particles (TiC and TiB2) used?

3) Figure 1a is not clear at all.

4) Provide EDS data for Figure 2.

5) The microstructures in Figure 3 are difficult to compare due to the different scale.

6) There is no standard deviation or t-statistic for the hardness values in Table 1. Therefore, it is not possible to evaluate the results statistically. Two decimal places in hardness values seem out of place.

7) Explain, is Figure 6 a calculation or an experiment? If it is experimental data, a description of the method is required.

8) There are many typos in the article. In particular, a lowercase letter is used at the beginning of many sentences (lines 14, 158, 185, and etc.).

9) The references are sloppy. Uniform style required.

Other points:

- Line 15-19. These two phrases need to be combined, because they are similar.

- Line 40. What is the Al3M phase?

- Line 42. The ACMZ alloy abbreviation should be explained at the first mention.

- Lines 67-68. The authors write: ‘As shown in Fig.1.(c), the ACMZ strip is rolled with a rolling amount 67 of 50% then isothermal holding for 6h at 673K’. In fact, there are no degrees of stain or exposure times in the Figure 1 (c).

- Line 73. What is the ‘hardness sample’ ?

Author Response

Point 1: The article is submitted as an original scientific manuscript, but its length is similar to a short communication. Therefore, a much more detailed description of the methods and results/discussion is required.

Response 1: Thank you for your valuable advice. As a non-native English graduate student, I try my best to explain the article clearly with my knowledge. Of course, it may still not be perfect, but please believe that I am trying my best to explain my experiments and results, and have supplemented the parts that are not detailed enough.

Point 2: Why are two types of nanosized particles (TiC and TiB2) used?

Response 2: In the previous quotation, it is mentioned that predecessors used two kinds of particles and achieved good results, and for our laboratory, the master alloy containing two kinds of particles can be easily purchased, so we chose two kinds of nanoparticles

Point 3: Figure 1a is not clear at all.

Response 3: For figure 1a, there is a problem with my statement. It is actually a cross-sectional process of alloy sheets produced by asymmetric TRC instrument.

Point 4: Provide EDS data for Figure 2.

Response 4: The purpose of Figure 2 is to prove that the microstructure and morphology of different positions in the alloy are different. Due to the simple alloy composition, my team and I agreed that it was not necessary to show EDS data.

Point 5: The microstructures in Figure 3 are difficult to compare due to the different scale.

Response 5: We have chosen a picture with a more suitable size.

Point 6: There is no standard deviation or t-statistic for the hardness values in Table 1. Therefore, it is not possible to evaluate the results statistically. Two decimal places in hardness values seem out of place.

Response 6: The data in Table 1 is already the calculated average. The calculation results are retained to two decimal places, which have been corrected and indicated in the title.

Point 7: Explain, is Figure 6 a calculation or an experiment? If it is experimental data, a description of the method is required.

Response 7: Figure 6 shows the values after tensile test, and the tensile sample size and tensile rate have been supplemented.

Point 8: There are many typos in the article. In particular, a lowercase letter is used at the beginning of many sentences (lines 14, 158, 185, and etc.).

Response 8: These errors have been corrected.

Point 9: The references are sloppy. Uniform style required.

Response 9: The format of references is the citation format provided by Google academic website, some of which cite Chinese academic papers. Maybe I am inexperienced and haven't found some specific document format problems, or there are format problems in the process of Chinese document translation.

Other points: These questions have been supplemented in the article.

Round 2

Reviewer 2 Report

It is difficult to understand the authors reply (especially, response 6 and 8) and these points keep unclear.  The authors should consider additional experiments to support the conclusions. 

Reviewer 3 Report

I carefully read the revised manuscript and the authors' responses to the remarks. I have to conclude that not all of them have been properly addressed. My recommendation is that the manuscript should be seriously revised or re-submitted to another journal.

1) Point 1: The article is submitted as an original scientific manuscript, but its length is similar to a short communication. Therefore, a much more detailed description of the methods and results/discussion is required.

Response 1: Thank you for your valuable advice. As a non-native English graduate student, I try my best to explain the article clearly with my knowledge. Of course, it may still not be perfect, but please believe that I am trying my best to explain my experiments and results, and have supplemented the parts that are not detailed enough.

Comment: English doesn't matter here. My comment is that the results are not described/discussed properly.

2) Point 3: Figure 1a is not clear at all.

Response 3: For figure 1a, there is a problem with my statement. It is actually a cross-sectional process of alloy sheets produced by asymmetric TRC instrument.

Comment: This is very good, but the Figure did not get better. You need to make an explanation in the Figure.

3) Point 6: There is no standard deviation or t-statistic for the hardness values in Table 1. Therefore, it is not possible to evaluate the results statistically. Two decimal places in hardness values seem out of place.

Response 6: The data in Table 1 is already the calculated average. The calculation results are retained to two decimal places, which have been corrected and indicated in the title.

Comment: That is indeed the essence of my comment: it is not enough to give only averages. Standard deviation or t-statistic for the hardness values to be provided.

4) Point 9: The references are sloppy. Uniform style required.

Response 9: The format of references is the citation format provided by Google academic website, some of which cite Chinese academic papers. Maybe I am inexperienced and haven't found some specific document format problems, or there are format problems in the process of Chinese document translation.

Comment: You should refer to the instructions for authors. There is references format. You are not using the uniform format now. For example, Ref. [14] … 700 (2017) 42–48 and Ref. [17] … 2017, 707: 58-64, i.e., you make a different format for year, volume and pages. That's what I mean.